# Training for endovascular therapy of acute arterial disease and procedure-related complication: An extracorporeally-perfused human cadaver model study

**Viktor Hartung**[1], **Anne Marie Augustin**[1], **Jan-Peter Grunz**[1], **Henner Huflage**[1], **Jan-Lucca Hennes**[1], **Florian Kleefeldt**[2], **Süleyman Ergün**[2], **Dominik Peter**[3], **Sven Lichthardt**[3], **Thorsten Alexander Bley**[1], **Philipp Gruschwitz**[1]*

1 Department of Diagnostic and Interventional Radiology, University Hospital of Würzburg, Würzburg, Germany, 2 Institute of Anatomy and Cell Biology, University of Würzburg, Würzburg, Germany, 3 Department of General, Visceral, Transplant, Vascular, and Pediatric Surgery, University Hospital of Würzburg, Würzburg, Germany

* Gruschwitz_P@ukw.de

## Abstract

### Purpose

The aim of this study was to evaluate the usability of a recently developed extracorporeally-perfused cadaver model for training the angiographic management of acute arterial diseases and periprocedural complications that may occur during endovascular therapy of the lower extremity arterial runoff.

### Materials and methods

Continuous extracorporeal perfusion was established in three fresh-frozen body donors via inguinal and infragenicular access. Using digital subtraction angiography for guidance, both arterial embolization (e.g., embolization using coils, vascular plugs, particles, and liquid embolic agents) and endovascular recanalization procedures (e.g., manual aspiration or balloon-assisted embolectomy) as well as various embolism protection devices were tested. Furthermore, the management of complications during percutaneous transluminal angioplasty, such as vessel dissection and rupture, were exercised by implantation of endovascular dissection repair system or covered stents. Interventions were performed by two board-certified interventional radiologists and one resident with only limited angiographic experience.

### Results

Stable extracorporeal perfusion was successfully established on both thighs of all three body donors. Digital subtraction angiography could be performed reliably and resulted in realistic artery depiction. The model allowed for repeatable training of endovascular recanalization and arterial embolization procedures with typical tactile feedback in a controlled environment. Furthermore, the handling of more complex angiographic devices could be

**Data Availability Statement:** All relevant data are within the manuscript.

**Funding:** PG; Z-02CSP/18; Interdisciplinary Center for Clinical Research (IZKF) at the University of Würzburg; https://www.med.uni-wuerzburg.de/izkf/startseite/ JPG.; Z-3BC/02; Interdisciplinary Center for Clinical Research (IZKF) at the University of Würzburg; https://www.med.uni-wuerzburg.de/izkf/startseite/ The funders had no role in study design, data collection and analysis, decision to publish, or preparation of the manuscript. TAB and JPG received speaker honoraria from Siemens Healthcare GmbH outside of the presented work. The Department of Diagnostic and Interventional Radiology of the University Hospital Würzburg receives ongoing research funding from Siemens Healthcare GmbH; https://www.siemens-healthineers.com/en-us The funders had no role in study design, data collection and analysis, decision to publish, or preparation of the manuscript.

**Competing interests:** The authors have declared that no competing interests exist.

exercised. Whereas procedural success was be ascertained for most endovascular interventions, thrombectomies procedures were not feasible in some cases due to the lack of inherent coagulation.

## Conclusion

The presented perfusion model is suitable for practicing time-critical endovascular interventions in the lower extremity runoff under realistic but controlled conditions.

## Introduction

Digital subtraction angiography (DSA) is an established imaging procedure to guide endovascular therapy during the management of acute and chronic vascular pathologies [1]. Angiographic intervention allows both, endovascular recanalization and arterial embolization procedures. While the former are used to treat acute limb ischemia [2], occluding procedures are suitable to stop bleeding from parenchymal organs or after soft tissue trauma [3]. In many cases, successful endovascular therapy renders surgical interventions unnecessary or the perioperative risk and subsequent morbidity can at least be reduced before definitive surgical therapy [4]. Due to this, the number of interventionally treated hemorrhages has increased in recent years [5, 6] and the mortality has been substantially reduced [7].

Angiographic treatment of time-critical emergencies requires rapid decision-making as well as high levels of skill and experience. Therefore, the interventionalist needs profound knowledge of treatment options and continuous training of their application [8, 9]. In addition, there is a large number of different devices with variable handling, which are often used infrequently, resulting in a lack of expertise. Since emergent endovascular treatments are often requested outside of the regular working hours, i.e. during on-call duty or weekend shifts [10, 11], support from more experienced colleagues may not be available all the time. Despite an increasing number of simulation courses, training for these time-critical scenarios is still largely limited to actual patient treatment under supervision. However, more in-depth training before a real-life endovascular procedure would be desirable.

Mock circulatory loops allow for the training of catheter handling and hands-on exercises with other interventional devices. Some larger centers even offer virtual angiography training suites, which facilitate the simulation of various interventions, albeit with a limited degree of realism and practical applicability due to the absence of human tissue properties and blood flow [12]. Both of these aspects have recently been addressed in a human cadaveric model with continuous extracorporeal perfusion [13]. While this model has previously been used for image quality comparisons between CT angiography scan protocol, its application for endovascular interventional training procedures has not been investigated yet.

Therefore, the aim of this study was to evaluate the perfusion model's usability for the training of DSA-guided management of vascular emergencies and periprocedural complications in a realistic but controlled environment.

## Materials and methods

### Cadaveric specimens and perfusion model

Three fresh-frozen cadavers were obtained from the local anatomical institute in accordance with national and European law (fully and irreversible anonymized human biomaterials).

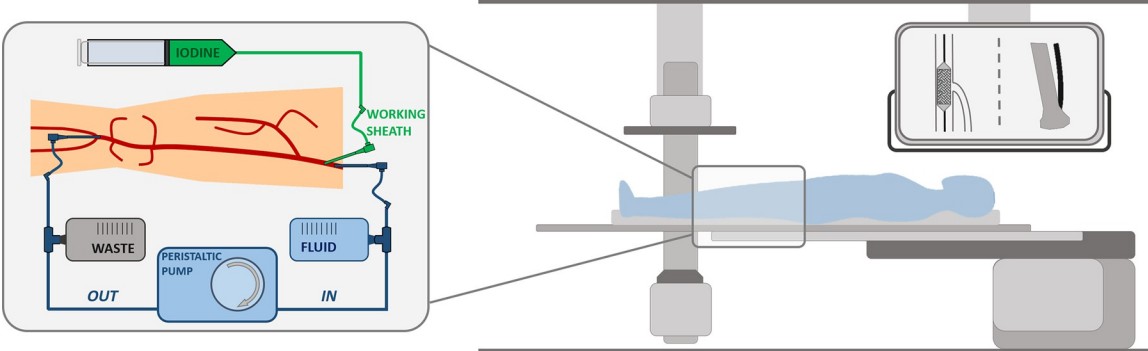

**Fig 1. Schematic of the experimental setup.** <u>Right</u>: Body donor in the angiography suite. <u>Left</u>: Perfusion circuit. The fluid reservoir (**FLUID**) compensates the loss of perfusion fluid during the examination. To perform digital subtraction angiography, contrast medium (**IODINE**) is manually injected via the working sheath and the resulting iodinated perfusion fluid is drained in a bag (**WASTE**). The **WORKING SHEATH** is used as vascular access to perform interventions.

Donors had authorized the usage of their bodies for education and research purposes with written informed consent during their lifetime (responsibility and accountability by the anatomical institute). The study received permission from the local institutional review board (Ethic Committee of the University of Würzburg; protocol number: 20220413 01).

After thawing of the non-chemically preserved cadavers, accesses to the common femoral artery (inguinal) and to the P3 segment of the popliteal artery (infragenicular) were surgically established. To achieve this, the vessels were sectioned at the level of the junction with the external iliac artery and the tibiofibular trunk, respectively. Remaining thrombotic material was removed using a Fogarty catheter. Subsequently, introducer sheaths (Flexor® sheath 8/6 French; Cook Medical, Bloomington, IN, USA) were inserted and fixated. The superficial femoral artery (SFA) and popliteal artery were subsequently perfused with a cooled mixture of Ringer's solution and 20% glucose solution (50:50) by means of a peristaltic pump. A schematic of the experimental setup is shown in Fig 1. Detailed explanations of the preparatory work and the materials required can be found elsewhere [13].

## Digital subtraction angiography

The cadavers were moved to the angiography suite and sterilely draped. Under continuous perfusion, a working sheath was inserted using the Seldinger technique and afterwards transfemoral antegrade fluoroscopic series were acquired in posterior-anterior orientation after manual injection of diluted contrast medium (50:50 mixture of Imeron® 350 mg/ml iodine and 0.9% sodium chloride solution). DSA was performed using a commercially available flat detector unit (Azurion 7 C20, Philips Healthcare, Best, The Netherlands) with vendor-recommended settings for low-dose examinations and a frame rate of 1–2 images/second.

## Interventional procedures

A representative selection of endovascular recanalization and arterial embolization procedures was conducted using the aforementioned *in-vitro* model. Investigated endovascular recanalization procedures included manual and balloon-assisted aspiration embolectomy. In addition, insertion and retrieval of an embolic protection device with a Nitinol "basket" was performed. In order to train vascular occlusion procedures, small muscle branches were intentionally perforated to simulate post-traumatic soft tissue bleeding or focal hemorrhages of parenchymal organs. After selective catheterization, afferent vessel segments were occluded using coils,

particle embolization, and/or liquid embolic agents. Treatment strategies for larger vessel injuries, e.g., in the context of a pelvic fracture, were also trained, including the deployment of a vascular plaque for artery occlusion or a covered stent to treat a previously induced hemorrhage. Finally, the management of periinterventional complications was trained. In addition to the above-mentioned treatment of a vessel rupture via a covered stent, the focus was on the therapy of dissections by means of ordinary vascular stents and special dissection repair systems. A summary of the performed endovascular interventions is given in Table 1.

## Results

### Digital subtraction angiography

Puncture of the femoral artery and insertion of a working sheath could by practiced in all cadaveric specimens. Correct intra-arterial needle position was indicated by realistic backflow of perfusion fluid. It may be noted that the femoral artery is uncovered after preparation and therefore easy to detect even with weak perfusion pressure. Manual DSA was performed as usual and appeared visually realistic. Smaller peripheral muscle branches were sufficiently contrasted (e.g., see Fig 2B).

### Arterial embolization procedures

A muscle branch of the SFA was catheterized using a selective microcatheter (Progreat[TM], 2.7F; Terumo, Tokyo, Japan). Of note, practicing the handling and positioning of this catheter is also relevant for the management of hemorrhages in parenchymatous organs. Afterwards, a hemorrhage was intentionally caused using a high tip load 0.014" wire (Astato XS; Asahi Intecc, Aichi, Japan) with subsequent placement of vascular coils (Concerto[TM] Detachable Coil System, Helix 4 mm x 8 cm; Medtronic, Minneapolis, MN, USA) to stop the contrast extravasation (Fig 2). In the DSA control, the vascular branch distal of the coils was mostly occluded.

Two other hemorrhages caused in the same manner were also occluded successfully after selective catheterization using combinations of coils (AZUR[™] CX, 3 mm x 8 cm, Terumo), liquid embolic agents (Histoacryl[®]; B. Braun, Melsungen, Germany) as well as pushable coils (Nester[®] Embolization Coils, 3 mm x 7 cm; Cook Medical) and tri-acryl gelatin microspheres (Embosphere[®] microspheres, 700–900 μm; Merit Medical Systems, South Jordan, UT, USA) as alternative therapeutic options. Images of both techniques are shown in Fig 3.

To simulate the treatment of larger vessel hemorrhages, e.g., of the internal iliac artery, a vascular plaque (Amplatzer[TM] Vascular Plaque II, 6 mm; Abbott Laboratories, North Chicago, IL, USA) was placed in the proximal SFA, repositioned several times for training purposes, and released after corrected positioning (Fig 4).

In another specimen, a vascular transection, which can occur periinterventionally or be of traumatic origin, was simulated. For this purpose, an oversized cutting balloon (Cutting Balloon Wolverine[TM], 2/6 mm; Boston Scientific, Marlborough, MA, USA) was expanded in the middle third of the SFA and the injured vessel segment was intentionally distended using an even more oversized angioplasty balloon (Atlas[TM] Gold, 18/20 mm; Becton Dickinson, Franklin Lakes, NJ, USA) until major vessel injury occurred. The rupture was already visible in DSA while the balloon was still expanded. Treatment was performed with a self-expanding covered stent (Covera[TM], 7/60 mm; Becton Dickinson). Subsequent DSA showed complete exclusion of the rupture, while remaining perivascular contrast agent was visible in the control angiogram (Fig 5).

**Table 1. Overview of endovascular interventions.**

| Type | Procedures | Material | Manufacturer | Comment/Limitation |
|---|---|---|---|---|
| arterial embolization | coil embolization (solo) | Concerto™ Detachable Coil System, Helix 4 mm x 8 cm | Medtronic, Minneapolis, MN, USA | almost complete occlusion despite lack of clotting |
| | coil embolization + liquid embolic | AZUR™ CX, 3mm x 8cm, Histoacryl® tissue adhesive | Terumo, Tokyo, Japan B. Braun, Melsungen, Germany | almost complete occlusion despite lack of clotting |
| | coil embolization + particle embolization | Nester® Embolization Coils, 3 mm x 7 cm Embosphere® microspheres, 700–900 µm | Cook Medical, Bloomington, IN, USA Merit Medical Systems, South Jordan, UT, USA | almost complete occlusion despite lack of clotting |
| | vascular plaque | Amplatzer™ Vascular Plaque II, 6 mm | Abbott Laboratories, North Chicago, IL, USA | only visual flow occlusion, visual control of placement |
| | covered stent occlusion | Covera™, 7/60 mm | Becton Dickinson, Franklin Lakes, NJ, USA | minor residual leakage hardly detectable due to remaining extravasation of dye |
| | temporary balloon occlusion | Atlas™ Gold, 18/20 mm | Becton Dickinson, Franklin Lakes, NJ, USA | (provoked arterial rupture) |
| endovascular recanalization | aspiration embolectomy (manual —catheter based) | BigLumen Aspiration Catheter, straight, 6F | OptiMed, Mannheim, Germany | artificial embolus (gelatine sponge) |
| | balloon embolectomy | Syntel™, 4F 80 cm, 0.8 ml balloon | LeMaitre Vascular Inc., Burlington, MA, USA | |
| embolism protection | Nitinol "basket" filter | SpiderFX™, 6 mm | Medtronic, Minneapolis, MN, USA | |
| dissection management | self-expanding stents | E-Luminexx™, 8/30 mm AbsolutePro™, 9/80 mm | Becton Dickinson, Franklin Lakes, NJ, USA Abbott Laboratories, North Chicago, IL, USA | in principle; no dissection present; practice correct placement and release of different models |
| | balloon-expanding stent | Herkulink Elite™, 7/18 mm | Abbott Laboratories, North Chicago, IL, USA | in principle; no dissection present |
| | endovascular dissection repair system | Tack® 6F, 6 x 8 mm | Philips Healthcare, Best, The Netherlands | in principle; no dissection present |
| | multiple stent system | VascuFlex® Multi-LOC, 6 x 13 mm | B. Braun, Melsungen, Germany | in principle; no dissection present |

## Endovascular recanalization procedures

For manual aspiration embolectomy, a 6F aspiration catheter (BigLumen Aspiration Catheter, straight, 6F; OptiMed, Mannheim, Germany) was inserted. An artificial occlusion simulated

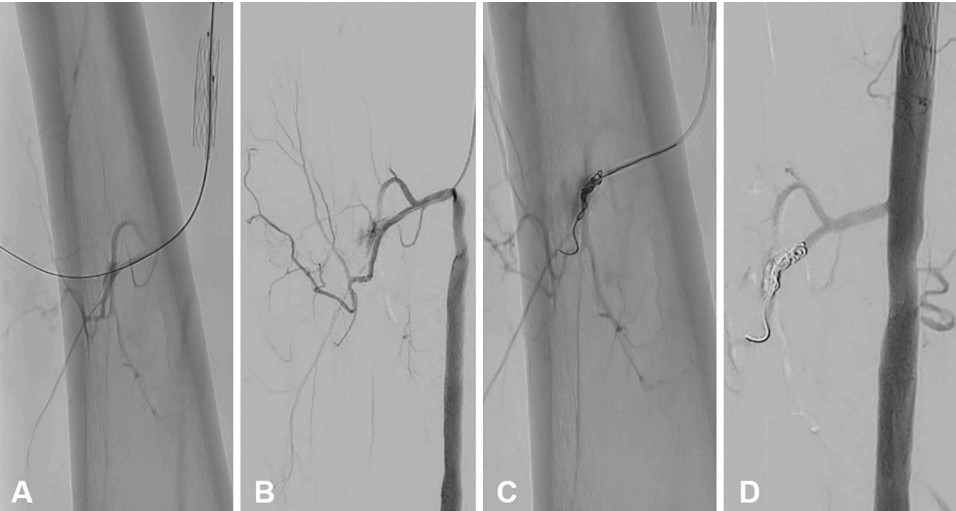

**Fig 2. Catheterization and treatment of a small vessel hemorrhage by coil embolization. A** Intentional penetration of a perforator vessel. **B** Visualization of hemorrhage using digital subtraction angiography. **C** Deposition of detachable coils (Concerto™, Helix 4 mm x 8 cm; Medtronic). **D** Occlusion of contrast extravasation after coil embolization.

with hemostatic absorbable gelatine sponge material (CuraSpon®; Cura Medical, Assendelft, The Netherlands) was aspirated and retrieved. After complete retrieval, DSA showed an unremarkable vascular runoff (Fig 6). In addition to manual aspiration embolectomy, a balloon-assisted mechanical embolectomy using a silicone embolectomy catheter (Syntel™, 4F 80 cm, 0.8 ml balloon; LeMaitre Vascular Inc., Burlington, MA, USA) was performed.

To train the application of periprocedural embolic prophylaxis, the insertion and retrieval of an embolic protection device comprised of a Nitinol "basket" (SpiderFX™, 6 mm; Medtronic) was tested. This tool prevents thrombotic material loosened during thrombectomy/embolectomy (or atherectomy) and can be retrieved together with the captured emboli (Fig 7).

## Dissection management

Dissections can occur after angioplasty and immediate management is often mandatory to maintain vessel patency. Three different options for dissection management were practiced: First, the established scaffolding technique is characterized by the deployment of stents adapted to the dissection length. The exact stent placement was trained by releasing a selection of stents from different manufacturers (E-Luminexx™, 8/30 mm; Becton Dickinson | Herkulink Elite™, 7/18 mm; Abbott | AbsolutePro™, 9/80 mm; Abbott). Second, a dedicated endovascular dissection repair system (Tack® 6F, 6 x 8 mm; Philips Healthcare) was used to treat another artificially-induced superficial femoral artery dissection. Third, multi-stent systems comprise several shorter stents with lower radial force. In this study, a system for spot stenting (VascuFlex® Multi-LOC, 6 x 13 mm; B. Braun) was tested in the cadaveric perfusion model (Fig 8).

## Discussion

The recently developed, continuous extracorporeally perfused human cadaver model allows for reliable establishment of vascular access and execution of digital subtraction angiographies. It enables the training of endovascular recanalization and arterial embolization procedures as well as practicing periprocedural complication management such as dissection and rupture

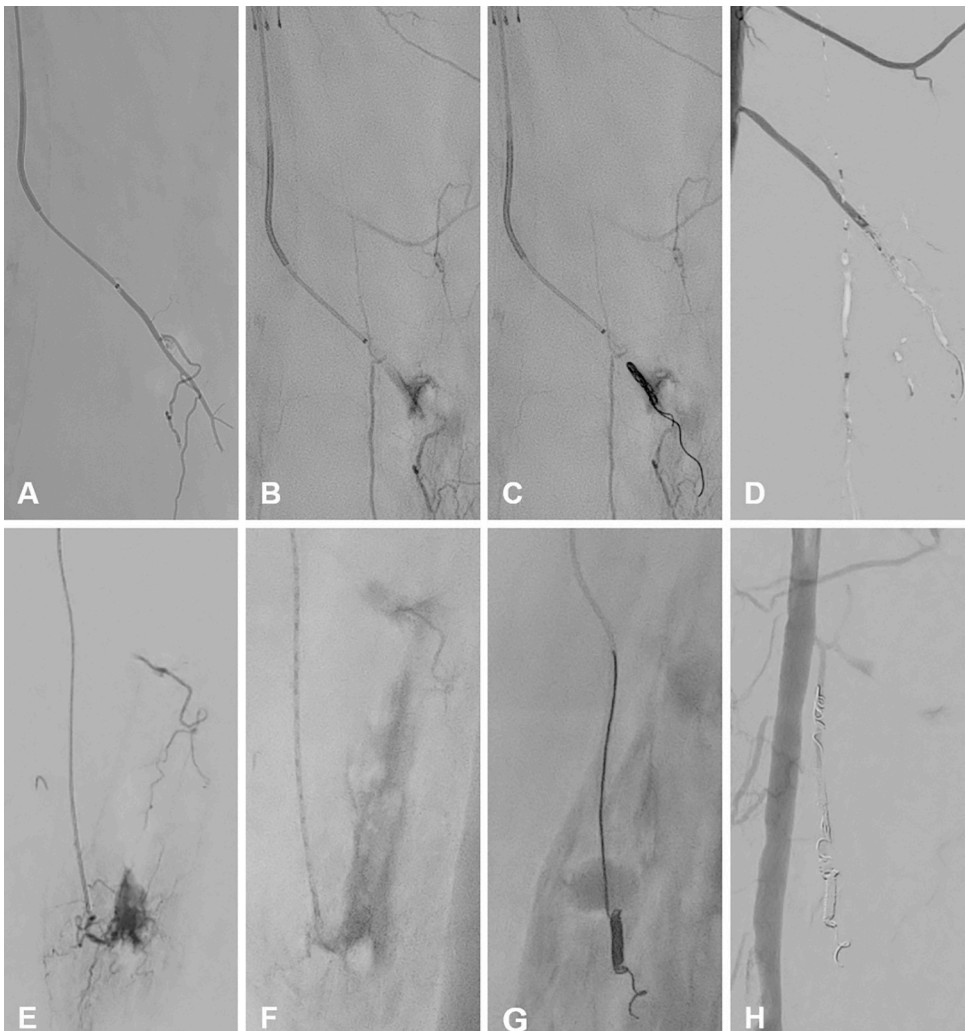

**Fig 3. Treatment of a small vessel hemorrhage by combinations of coil, liquid tissue adhesive, and particle embolization. A** Selective catheterization of a muscle branch in the upper leg. **B** Visualization of the induced hemorrhage in DSA. **C** Deposited coils (AZUR™ CX, 3mm x 8cm, Terumo) in the perforated vessel. **D** Stagnant bleeding in DSA after additional application of liquid embolic (Histoacryl®; B. Braun) **E** Another intentionally caused bleeding visualized over a selective catheter (Progreat™; Terumo). **F** Fractional application of particle embolisate (Embosphere® microspheres, 700–900 μm; Merit Medical) visible as contrast defects inside the catheter. **G** Additional deposition of two coils (Nester® Embolization Coils, 3 mm x 7 cm; Cook Medical). **H** Cessation of bleeding in the DSA control.

repair under realistic conditions with characteristic tactile feedback using the real angiographic equipment.

The frequency of endovascular and thus minimally invasive treatment procedures have steadily increased in recent years [5, 6]. In Germany, over 300.000 percutaneous interventions were conducted in 2021 with numbers doubled from 2005 [14]. Of these, a substantial percentage comprise emergency procedures in patients with acute, potentially life-threatening conditions. For example, every tenth to every third polytraumatized patient presents with a peripheral vascular injury [15, 16]. While endovascular therapy procedures have been shown to be effective alternatives to open surgical treatment [7], the pressure to provide an appropriate expertise around the clock has markedly increased. In addition to growing numbers of

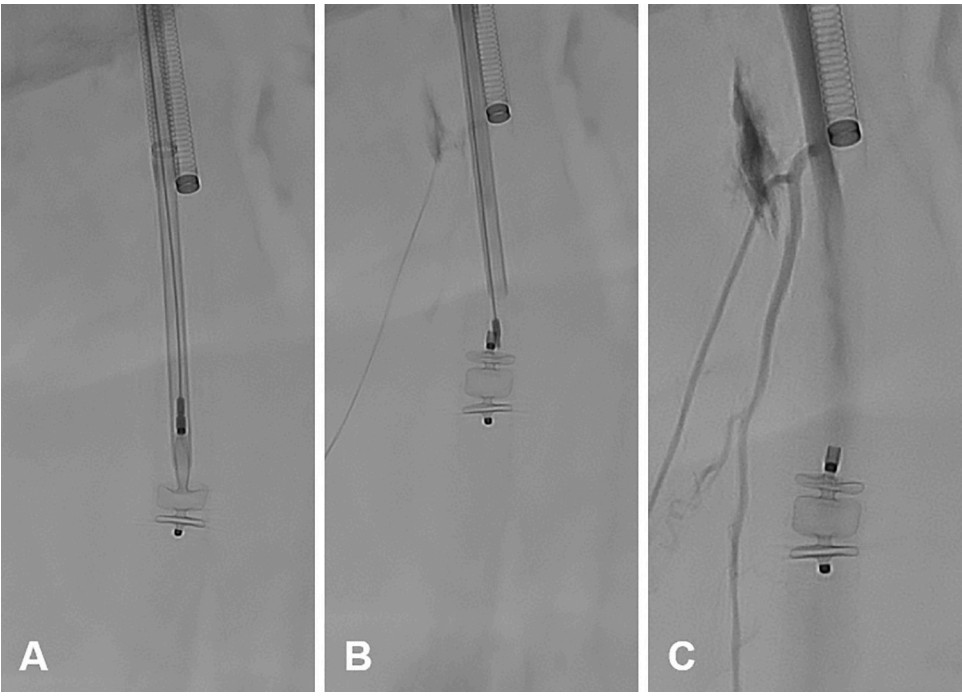

**Fig 4. Placement and release of a vascular plaque. A** Insertion of a vascular plaque (Amplatzer^TM Vascular Plaque II, 6 mm; Abbott) via the working sheath. **B** Release of the plaque by turning the thread. **C** Vascular plaque in intravascular position. Note: Circumscribed hemorrhage adjacent to the inflow sheath after preparation. Working sheath and inflow sheath aligned parallelly.

interventions outside regular working hours [10, 11, 17], this also increases the need for appropriately specialized interventionalists as well as in-depth training in the angiographic skills required for such procedures.

Besides supervised learning from experienced colleagues during actual procedures, which is generally associated with a non-negligible risk for the patient [8], model-based learning

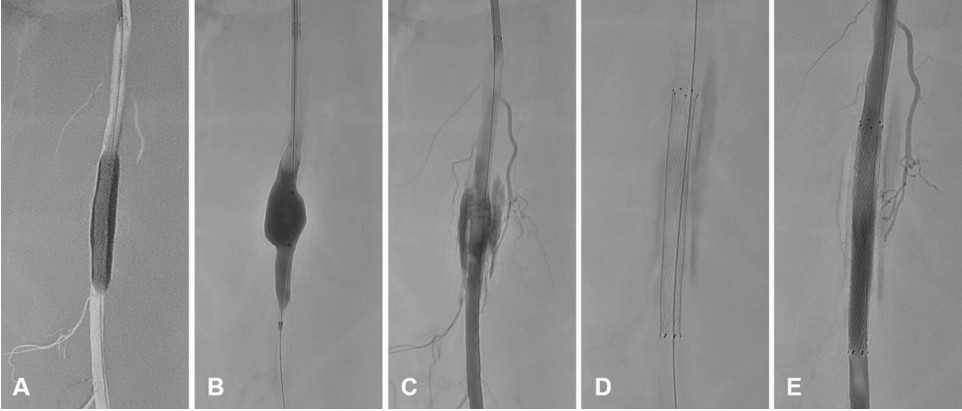

**Fig 5. Treatment of a simulated vessel rupture using a covered stent. A** Intentional injury of the vessel wall by expanding an oversized cutting balloon (Cutting Balloon Wolverine^TM, 2/6 mm; Boston Scientific). **B** Vessel rupture induced via balloon angioplasty (Atlas^TM Gold, 18/20 mm; Becton Dickinson). **C** Verification of major hemorrhage after balloon deflation. **D** Implantation of a covered stent (Covera^TM, 7/60 mm; Becton Dickinson). **E** Successful exclusion of the vascular transection.

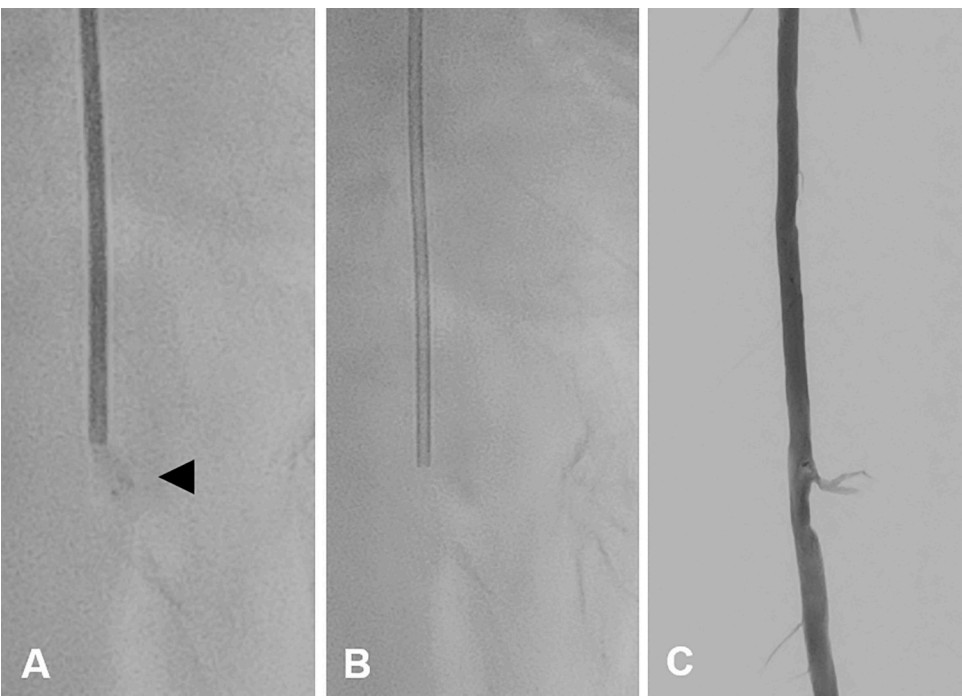

**Fig 6. Manual aspiration embolectomy of artificial embolus. A** Aspiration catheter (BigLumen Aspiration Catheter, straight, 6F; OptiMed) filled with contrast agent for improved visibility. Note the attached artificial embolus (CuraSpon®; Cura Medical) marked with an arrowhead. **B** Empty aspiration catheter. **C** Unremarkable vessel delineation after the artificial embolus was successfully retrieved.

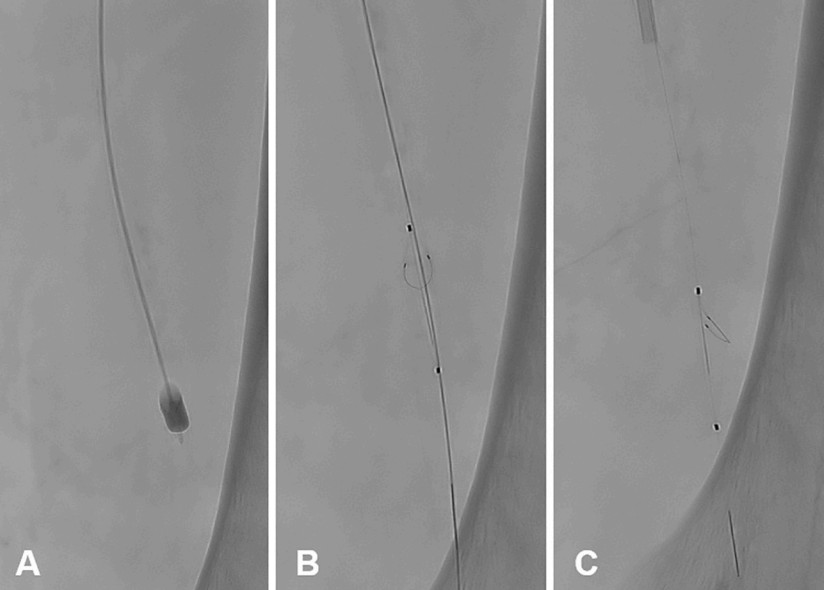

**Fig 7. Balloon-assisted embolectomy and embolic protection. A** Silicone balloon embolectomy catheter (Synthel™, 4F 80 cm, 0.8 ml; LeMaitre) in intravascular position **B/C** Embolic protection device with Nitinol "basket" (SpiderFX™, 6 mm; Medtronic) depicted in two different angulations.

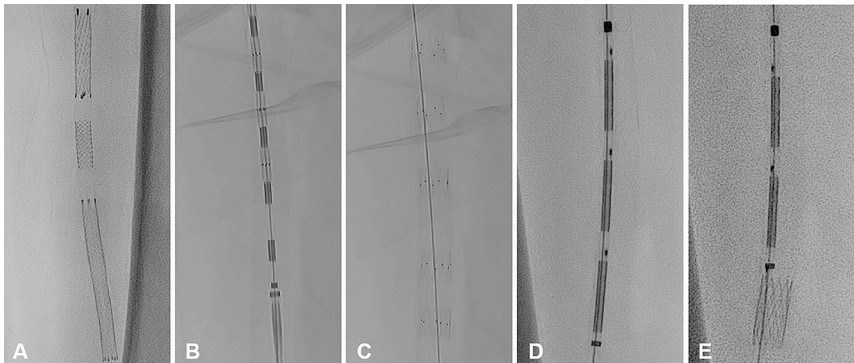

**Fig 8. Variations of vascular dissection treatment options. A** Intravascular stent positioning (top: E-Luminexx™, 8/
30 mm; Becton Dickinson | middle: Herkulink Elite™, 7/18 mm; Abbott | bottom: AbsolutePro™, 9/80 mm; Abbott).
**B/C** Endovascular dissection repair system (Tack® 6F, 6 x 8 mm; Philips) before (B) and after (C) stent release **D/E**
Multiple stent system (VascuFlex® Multi-LOC, 6 x 13 mm; B. Braun) before (D) and after (E) stent release.

approaches have emerged as a potential alternative in recent years [18]. Expectedly, multiple
studies have shown that training generally contributes to a reduction in patient morbidity and
mortality from interventional procedures [19–22]. Hseino et al. [23] and Patel et al. [24],
among others, demonstrated a positive learning curve for model-based training of procedures
such as treatment of the aorta, renal arteries, the carotid artery, and intracranial vessels. In
addition, dedicated exercises reduced the total treatment and fluoroscopy time [25, 26] with
less contrast agent administered during the procedures [24, 25, 27, 28]. Apart from objective
criteria, a subjective increase in knowledge was also demonstrated in self-assessment assays
[18]. Model-based training allows for controlled honing of skills [29, 30] without a risk to
patients [31]. In particular, the option to perform interventions in a protected environment
under supervision [32] with frequent repetition of individual steps leads to a reduction of peri-
procedural errors [18]. While evidence shows that the translation of skills to patient care is
generally successful [23, 33], it must be stated that model-based learning can only be a supple-
ment for actual training in real-world patients [34, 35].

Prior studies indicate that both novices [23, 25, 36–38] and more experienced intervention-
alists [27, 39] benefit from training on models. However, models must be adapted to the
appropriate level of knowledge or skills and increase in complexity. While simple synthetic
models are sufficient for beginners to learn how to use a guide wire or catheter, more refined
models are required for complex procedures. The use of large animal models, while anatomi-
cally realistic, requires expensive resources such as animal housing and laboratory [33, 40].
Moreover, Berry et al. could show that the translation of skills from animal models is worse
compared to virtual angiography simulations [33]. Finally, there is ongoing debate on the ethi-
cal adequacy of large animal models, since alternatives are available [41].

Currently, the best training options are arguably endovascular simulators and models based
on human body donors [12]. Virtual simulators are comparatively expensive to set up, require
maintenance, and may have recurring costs like software licenses and support. In turn, exami-
nations can be repeated infinitely and the range of interventions that can be simulated is wide.
Since no x-ray exposure is required, protective garment such as lead aprons can be omitted.
However, while simulators only generate virtual catheters, balloons, and stents, human cadaver
models can be used to train the handling of actual devices, some of which are difficult to oper-
ate. This also applies to procedures with increased complication rates, such as particle or coil
embolization, the use of liquid embolic agent, mechanical embolectomy systems or the use of

closure systems, among others. Contrary to models described in the literature [28, 42, 43], our cadaveric model provides a continuous perfusion circuit with the option of realistic DSA of the peripheral arterial runoff in a real-life angiography suite. Various angiographic procedures can be trained in a controlled environment with an entire team to reinforce and deepen work-flows, which has been shown to improve teamwork and therefore patient care [44].

Several limitations of the proposed model have to be mentioned. First, cadavers are a resource with limited availability imposing recurring costs as opposed to the one-time pur-chase of a simulator. This availability problem can primarily be solved through close coopera-tion with the providing institutes, and the possibility of multiple use of a single donor for different procedures is particularly valuable. In the long term, the shortage of suitable body donors can probably only be overcome by raising awareness and thus increasing the willing-ness to donate. Second, assistance from surgically experienced colleagues is required for the preparation of vascular access and only non-preserved fresh-frozen cadavers are suitable for establishment of the model. After thawing, these can be used for a maximum of 48 hours, hence only a limited number of procedures per donor is feasible, especially if vascular injuries are provoked and/or stents are inserted. Since the perfusion fluid has no intrinsic coagulation, efficiency of pro-thrombotic procedures is limited. Third, in this feasibility study, endovascu-lar interventions were only performed on three body donors. It may be noted, however, that the perfusion model has been successfully established in more than ten body donors in the meantime, suggesting good reproducibility. Fourth, a real educational situation was tested with one resident performing a coil embolization as a pilot trial. While no structured evalua-tion was conducted, further studies are planned to investigate the training potential in compar-ison to existing alternatives like virtual angiography suites. Finally, due to the effort and resources required to set up the continuous extracorporeal perfusion, we advocate that the model should be reserved for advanced training and practice of complex angiographic proce-dures and material.

## Conclusion

The presented perfusion model is suitable for practicing time-critical endovascular interven-tions in the lower extremity runoff under realistic but controlled conditions.

## Author Contributions

**Conceptualization:** Viktor Hartung, Anne Marie Augustin, Jan-Peter Grunz, Henner Huflage, Florian Kleefeldt, Süleyman Ergün, Thorsten Alexander Bley, Philipp Gruschwitz.

**Data curation:** Viktor Hartung, Anne Marie Augustin, Henner Huflage, Jan-Lucca Hennes, Philipp Gruschwitz.

**Formal analysis:** Jan-Lucca Hennes, Dominik Peter, Sven Lichthardt, Philipp Gruschwitz.

**Funding acquisition:** Jan-Peter Grunz, Thorsten Alexander Bley, Philipp Gruschwitz.

**Investigation:** Viktor Hartung, Anne Marie Augustin, Jan-Peter Grunz, Henner Huflage, Jan-Lucca Hennes, Florian Kleefeldt, Dominik Peter, Sven Lichthardt, Philipp Gruschwitz.

**Methodology:** Viktor Hartung, Anne Marie Augustin, Jan-Peter Grunz, Henner Huflage, Jan-Lucca Hennes, Süleyman Ergün, Dominik Peter, Sven Lichthardt, Philipp Gruschwitz.

**Project administration:** Viktor Hartung, Florian Kleefeldt, Süleyman Ergün, Thorsten Alex-ander Bley, Philipp Gruschwitz.

**Resources:** Henner Huflage, Florian Kleefeldt, Süleyman Ergün, Dominik Peter, Sven Lichthardt, Thorsten Alexander Bley.

**Software:** Henner Huflage, Thorsten Alexander Bley.

**Supervision:** Anne Marie Augustin, Jan-Peter Grunz, Jan-Lucca Hennes, Florian Kleefeldt, Thorsten Alexander Bley, Philipp Gruschwitz.

**Validation:** Viktor Hartung, Anne Marie Augustin, Jan-Peter Grunz, Henner Huflage, Jan-Lucca Hennes, Süleyman Ergün, Dominik Peter, Sven Lichthardt.

**Visualization:** Philipp Gruschwitz.

**Writing – original draft:** Viktor Hartung, Jan-Peter Grunz, Philipp Gruschwitz.

**Writing – review & editing:** Viktor Hartung, Anne Marie Augustin, Jan-Peter Grunz, Henner Huflage, Jan-Lucca Hennes, Florian Kleefeldt, Süleyman Ergün, Dominik Peter, Sven Lichthardt, Thorsten Alexander Bley, Philipp Gruschwitz.

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
