## [Decision Letter · Decision Letter 0]

31 Oct 2023

PONE-D-23-26109Acute pathology and complication management in endovascular therapy of the lower extremity: Practicing time-critical interventions on an extracorporeally-perfused human cadaver modelPLOS ONE

Dear Dr. Gruschwitz,

Thank you for submitting your manuscript to PLOS ONE. After careful consideration, we feel that it has merit but does not fully meet PLOS ONE’s publication criteria as it currently stands. Therefore, we invite you to submit a revised version of the manuscript that addresses the points raised during the review process.

We look forward to receiving your revised manuscript.

Kind regards,

Ezio Lanza, M.D.

Academic Editor

PLOS ONE

Journal Requirements:

4. We note that Figure 1 in your submission contain copyrighted images. All PLOS content is published under the Creative Commons Attribution License (CC BY 4.0), which means that the manuscript, images, and Supporting Information files will be freely available online, and any third party is permitted to access, download, copy, distribute, and use these materials in any way, even commercially, with proper attribution. For more information, see our copyright guidelines: http://journals.plos.org/plosone/s/licenses-and-copyright.

Reviewers' comments:

Reviewer's Responses to Questions

**Comments to the Author**

1. Is the manuscript technically sound, and do the data support the conclusions?

Reviewer #1: Yes

Reviewer #2: Yes

2. Has the statistical analysis been performed appropriately and rigorously? 

Reviewer #1: N/A

Reviewer #2: N/A

3. Have the authors made all data underlying the findings in their manuscript fully available?

Reviewer #1: Yes

Reviewer #2: Yes

4. Is the manuscript presented in an intelligible fashion and written in standard English?

Reviewer #1: Yes

Reviewer #2: Yes

5. Review Comments to the Author

Reviewer #1: 1. The title seems in need of correction.

Since “pathology” has a strong pathological feel, it would be better to change it to “acute arterial disease.” It would be good to change “complication” to “procedure-related complication” or periprocedural complication.

Since the title is too long, I think it could be “Training for endovascular therapy of acute arterial disease and procedure-related complication: An extracorporeally-perfused human cadaval model study.”

2. Line 4; Please also edit “pathologies and complications” in the 4th line to fit the title.

3. Line 5; I think “arterial” would be better instead of “vascular.”

4. Line 8; I think "arterial embolization" is a better word for this topic than "vessel-occluding".

Please make the same corrections to lines 19, 34, 101, 105, 110, 127, 228 and the table.

5. Line 10; I think it would be better to use “endovascular recanalization” rather than “vessel-opening”.

Regarding this, please make the same corrections to lines 19, 34, 101, 102, 183, 228 and the table.

6. Line 13; Please also add “endovascular dissection repair system” before “covered stent.”

7. Line 19, 230; “Haptics” refers to a technology that generates vibration, force, or shock in various digital devices to give users a sense of touch. Since it does not fit with the main text, it would be better to delete it or edit it with other words.

8. Line 22; Rather than simply writing “intervention,” it would be better to change it to “endovascular procedure” or “endovascular intervention.”

9. Line 22; Please change it to “thrombectomies” rather than “thrombosis-inducing procedures.”

10. Line 35; I think it would be better to express it as “acute limb ischemia” rather than “acute ischemia of the vascular periphery.”

11. Line 46-47; Rather than “emergency intervention,” “emergent endovascular treatments” seems to be a more appropriate term for the paper.

12. Line 53; I think “Mock circulatory loops” is a more appropriate term for the paper than “Simplified fluid mock models.”

13. Line 79; Please write the product name and manufacturer of the sheath used.

14. Line 152; Please change “liquid embolic” to “liquid embolic material” or “liquid embolic agent.”

15. Line 186; Since this is an American paper, please change “heamostatic” to “hemostatic.”

16. Figure 8; To match the order of product description with the main text, the order of Figures B/C and Figures D/E must be changed.

17. Line 229; It would be good to change “complication” to “procedure-related complication” or periprocedural complication.

18. Line 252; Please change “intervention” to “procedure” or “treatment.”

19. Line 280; Please change “liquid tissue adhesive” to “liquid embolic agent.”

20. Line 295; Rather than simply writing “intervention,” it would be better to change it to “endovascular procedure” or “endovascular intervention.”

21. Were the products used in the procedure purchased with research funds or donated by the manufacturer?

Reviewer #2: Nice paper and idea is quite interesting.

The cadaver training is well established in open surgery particularly in vessel exposure and dissection.

There is no much mentioned in literature about endovascular intervention using cadavers.

I still believe that it will not be the same as interventions in real life as different settings and different pathologies particularly hard calcification angulated vessels. However, doing this training can get the operators to become familiar with different techniques and instruments.

Another limitation is the availability of such suitable cadavers that the author should address that and how to overcome this problem.

6. PLOS authors have the option to publish the peer review history of their article (what does this mean?). If published, this will include your full peer review and any attached files.

Reviewer #1: No

Reviewer #2: **Yes: **Baker Ghoneim

---

## [Author Response · Author response to Decision Letter 0]

21 Nov 2023

We would like to thank the reviewers for their efforts and suggestions. 

Below you will find the point-by-point responses to your comments. 

Reviewer #1: 

1. The title seems in need of correction.

Since “pathology” has a strong pathological feel, it would be better to change it to “acute arterial disease.” It would be good to change “complication” to “procedure-related complication” or periprocedural complication.

Since the title is too long, I think it could be “Training for endovascular therapy of acute arterial disease and procedure-related complication: An extracorporeally-perfused human cadaval model study.”

The title was changed according to the recommendation.

2. Line 4; Please also edit “pathologies and complications” in the 4th line to fit the title.

The line has been adjusted accordingly.

3. Line 5; I think “arterial” would be better instead of “vascular.”

The words were exchanged.

4. Line 8; I think "arterial embolization" is a better word for this topic than "vessel-occluding".

Please make the same corrections to lines 19, 34, 101, 105, 110, 127, 228 and the table.

The naming has been adjusted and subsequently amended.

5. Line 10; I think it would be better to use “endovascular recanalization” rather than “vessel-opening”.

Regarding this, please make the same corrections to lines 19, 34, 101, 102, 183, 228 and the table.

The naming has been adjusted and subsequently amended.

6. Line 13; Please also add “endovascular dissection repair system” before “covered stent.”

The phrase has been added.

7. Line 19, 230; “Haptics” refers to a technology that generates vibration, force, or shock in various digital devices to give users a sense of touch. Since it does not fit with the main text, it would be better to delete it or edit it with other words.

The term should refer to the sensory experience of the devices while usage, e.g. slipping of the catheter into a side branch. As this is a relevant advantage over other models, we have decided to retain the text passage and replace the word "haptic" with "tactile".

8. Line 22; Rather than simply writing “intervention,” it would be better to change it to “endovascular procedure” or “endovascular intervention.”

The wording has been changed accordingly.

9. Line 22; Please change it to “thrombectomies” rather than “thrombosis-inducing procedures.”

The wording has been changed accordingly.

10. Line 35; I think it would be better to express it as “acute limb ischemia” rather than “acute ischemia of the vascular periphery.”

The wording has been changed accordingly.

11. Line 46-47; Rather than “emergency intervention,” “emergent endovascular treatments” seems to be a more appropriate term for the paper.

The wording has been changed accordingly.

12. Line 53; I think “Mock circulatory loops” is a more appropriate term for the paper than “Simplified fluid mock models.”

The term has been changed accordingly.

13. Line 79; Please write the product name and manufacturer of the sheath used.

The product name and manufacturer have been added.

14. Line 152; Please change “liquid embolic” to “liquid embolic material” or “liquid embolic agent.”

The wording has been changed accordingly.

15. Line 186; Since this is an American paper, please change “heamostatic” to “hemostatic.”

The spelling has been corrected.

16. Figure 8; To match the order of product description with the main text, the order of Figures B/C and Figures D/E must be changed.

The text passage has been adjusted and is now in the same order as the figure.

17. Line 229; It would be good to change “complication” to “procedure-related complication” or periprocedural complication. 

The wording has been changed accordingly.

18. Line 252; Please change “intervention” to “procedure” or “treatment.”

The wording has been changed accordingly.

19. Line 280; Please change “liquid tissue adhesive” to “liquid embolic agent.”

The wording has been changed accordingly.

20. Line 295; Rather than simply writing “intervention,” it would be better to change it to “endovascular procedure” or “endovascular intervention.”

The wording has been changed accordingly.

21. Were the products used in the procedure purchased with research funds or donated by the manufacturer?

The materials and devices used had passed their expiration date and were no longer suitable for patient care for hygienic reasons but were kept for demonstration and teaching purposes and used in this task for this study.

Reviewer #2: 

Nice paper and idea is quite interesting.

The cadaver training is well established in open surgery particularly in vessel exposure and dissection.

There is no much mentioned in literature about endovascular intervention using cadavers.

I still believe that it will not be the same as interventions in real life as different settings and different pathologies particularly hard calcification angulated vessels. However, doing this training can get the operators to become familiar with different techniques and instruments.

Another limitation is the availability of such suitable cadavers that the author should address that and how to overcome this problem.

Thank you for the positive criticism. The fact of limited availability of body donors has been further elaboration in the “Limitation” section.

---

## [Editor Report · Decision Letter 1]

15 Jan 2024

Training for endovascular therapy of acute arterial disease and procedure-related complication: An extracorporeally-perfused human cadaver model study.

PONE-D-23-26109R1

Dear Dr. Gruschwitz,

We’re pleased to inform you that your manuscript has been judged scientifically suitable for publication and will be formally accepted for publication once it meets all outstanding technical requirements.

Kind regards,

Ezio Lanza, M.D.

Academic Editor

PLOS ONE

---

## [Editor Report · Acceptance letter]

31 Jan 2024

PONE-D-23-26109R1 

PLOS ONE

Dear Dr. Gruschwitz, 

I'm pleased to inform you that your manuscript has been deemed suitable for publication in PLOS ONE. Congratulations! Your manuscript is now being handed over to our production team.

Kind regards, 

on behalf of

Dr. Ezio Lanza 

Academic Editor

PLOS ONE